# Examining the Transcriptomic and Biochemical Signatures of *Bacillus subtilis* Strains: Impacts on Plant Growth and Abiotic Stress Tolerance

**DOI:** 10.3390/ijms241813720

**Published:** 2023-09-06

**Authors:** Peter E. Chang, Yun-Hsiang Wu, Ciao-Yun Tai, I-Hung Lin, Wen-Der Wang, Tong-Seung Tseng, Huey-wen Chuang

**Affiliations:** Department of Agricultural Biotechnology, National Chiayi University, Chiayi 600355, Taiwans1082391@mail.ncyu.edu.tw (C.-Y.T.); s1100111@mail.ncyu.edu.tw (I.-H.L.);

**Keywords:** transcription profiling, exopolysaccharide, ABA signaling, drought stress tolerance, heat stress tolerance, copper stress tolerance

## Abstract

Rhizobacteria from various ecological niches display variations in physiological characteristics. This study investigates the transcriptome profiling of two *Bacillus subtilis* strains, BsCP1 and BsPG1, each isolated from distinct environments. Gene expression linked to the synthesis of seven types of antibiotic compounds was detected in both BsCP1 and BsPG1 cultures. Among these, the genes associated with plipastatin synthesis were predominantly expressed in both bacterial strains. However, genes responsible for the synthesis of polyketide, subtilosin, and surfactin showed distinct transcriptional patterns. Additionally, genes involved in producing exopolysaccharides (EPS) showed higher expression levels in BsPG1 than in BsCP1. Consistently with this, a greater quantity of EPS was found in the BsPG1 culture compared to BsCP1. Both bacterial strains exhibited similar effects on *Arabidopsis* seedlings, promoting root branching and increasing seedling fresh weight. However, BsPG1 was a more potent enhancer of drought, heat, and copper stress tolerance than BsCP1. Treatment with BsPG1 had a greater impact on improving survival rates, increasing starch accumulation, and stabilizing chlorophyll content during the post-stress stage. qPCR analysis was used to measure transcriptional changes in *Arabidopsis* seedlings in response to BsCP1 and BsPG1 treatment. The results show that both bacterial strains had a similar impact on the expression of genes involved in the salicylic acid (SA) and jasmonic acid (JA) signaling pathways. Likewise, genes associated with stress response, root development, and disease resistance showed comparable responses to both bacterial strains. However, treatment with BsCP1 and BsPG1 induced distinct activation of genes associated with the ABA signaling pathway. The results of this study demonstrate that bacterial strains from different ecological environments have varying abilities to produce beneficial metabolites for plant growth. Apart from the SA and JA signaling pathways, ABA signaling triggered by PGPR bacterial strains could play a crucial role in building an effective resistance to various abiotic stresses in the plants they colonize.

## 1. Introduction

The microorganisms known as plant growth-promoting rhizobacteria (PGPR) are a diverse set of microorganisms that colonize the rhizosphere and stimulate plant growth via various mechanisms. Rhizobacteria produce diverse metabolites to function as communication signals within the microbial ecosystem. This may provide them an advantage in the competition for resources and habitats against other organisms. Certain metabolites from PGPR strains can have a direct effect on plant development. For example, the production of indole-3-acetic acid (IAA), a prevalent form of the phytohormone auxin, is commonly found among bacteria. IAA can serve a role in facilitating communication between cells within and among microbial communities, as well as in interactions between plants and microbes [1]. In plant cells, the regulatory role of auxin is evident throughout all stages of the plant life cycle. In particular, the essential role of auxin in lateral root development has been widely recognized [2]. IAA, produced by PGPR strains, positively influences root branching and vegetative growth [3]. Microbial bacillibactin, a catechol type siderophore, not only works as a biocontrol agent, but also promotes plant growth by improving iron absorption [4]. However, the metabolites produced by PGPR strains can also boost plant growth through indirect processes. For example, applying PGPR that produce 1-aminocyclopropane-1-carboxylate (ACC) deaminase promotes plant growth by mitigating the harmful effects of ethylene production under stressful conditions [5]. Rhizobacteria can produce various types of volatile compounds (VOCs) that not only facilitate interactions between microorganisms but also have the ability to modulate plant growth and stress tolerance by altering phytohormone signaling pathways [6,7,8,9]. Surfactin, a cyclic lipopeptide, plays multiple roles in microorganisms, including promoting biofilm formation, facilitating cell motility, and engaging in competition with other microorganisms [10]. In plant cells, surfactin can trigger early events that are associated with the induction of defense responses, such as extracellular alkalization, the accumulation of reactive oxygen species (ROS), and the activation of defense-related enzymes like lipoxygenase [11]. Exopolysaccharides (EPSs) are the main components for biofilm structure formation [12]. EPSs can act as chelating agents, limiting the uptake of heavy metals in plant roots and enhancing chlorophyll and osmolyte concentrations, thus mitigating cellular damage under drought stress [13,14].

The metabolites produced by PGPR can promote plant growth by altering multiple plant signaling pathways that are involved in the regulation of adaptive responses under stressful conditions [15]. Plants encounter various abiotic stresses that impose negative effects on their growth and productivity. During stages of water shortage, the closing of stomata is a strategy to prevent excessive water loss. However, this process also inhibits CO_2_ absorption in the leaves, leading to a reduction in photosynthesis. This in turn results in an accumulation of ROS, which could be detrimental to the photosystem and instigate chlorophyll degradation [16]. Heat stress disrupts protein function and results in metabolic irregularities. As a consequence, there is an increase in oxidative stress, which can have harmful impacts on plant growth [17]. Heavy metals have the potential to interrupt cellular activities by replacing essential metals and inducing protein misfolding and aggregation. These events can impact protein stability and decrease cell viability. Moreover, certain redox-reactive heavy metals, such as iron and copper, can produce ROS molecules through the Fenton reaction [18]. The build-up of ROS is a common factor contributing to cellular damage under various abiotic stresses. Consequently, the fitness of plants in stressful environments heavily relies on their antioxidant defense system, which consists of antioxidant enzymes and secondary metabolites possessing ROS scavenging activity [19]. Apart from ROS signaling, the signaling pathways of phytohormones play significant roles in the regulation of plant response to abiotic stress. For example, the expression of *9-cis-epoxycarotenoid dioxygenase 3* (*NCED3*), which encodes a rate-limiting enzyme for the synthesis of abscisic acid (ABA), is induced to higher levels in response to a water deficit signal. [20]. The synthesis of ABA prompts the activity of respiratory burst oxidase homologues (RBOHs), leading to the production of ROS. This results in the closure of stomata and the prevention of water loss [21]. Systemic immunity, triggered by microbe–plant interaction, encompasses both systemic acquired resistance (SAR) and induced systemic resistance (ISR). SAR is a long-lasting form of resistance against pathogens that requires salicylic acid (SA) as a signal molecule [22]. ISR is activated by microorganisms residing in the rhizosphere, with jasmonic acid (JA) acting as a signal molecule to activate this induced resistance [22]. In addition to their roles in induced disease resistance, recent studies also indicate an important role of SA and JA signaling in the regulation of abiotic stress tolerance through mediating antioxidant activity and osmolyte accumulation [23,24]. The antioxidant defense system has been reported to be activated by SA, which helps improve the function of photosystems under heat stress [25]. The application of SA can promote the accumulation of proline, which helps to counteract the osmotic stress generated in high-temperature environments and enhance heat stress tolerance [26]. Similarly, it has been reported that exogenous JA can activate the antioxidant defense system, modify osmotic adjustment, and improve heat stress tolerance [27].

The metabolites of PGPR strains can positively regulate plant growth by modulating phytohormone signaling pathways linked to plant development. Research has indicated that microbial VOCs can induce systemic resistance against both abiotic and biotic stress by altering the SA and ABA signaling pathways [6,7,8]. A primary constituent of VOCs produced by rhizobacteria, 2,3-Butanediol (2,3-BD), has been shown to enhance disease resistance and activate the auxin signal in plants when applied [9,28]. Surfactin triggers cellular pathways linked to SA and JA signaling to induce disease resistance in wheat against pathogen attacks [29]. Rhizobacteria that produce polyamines, components of biofilms, have been found to activate the ABA signaling pathway, thereby enhancing drought stress tolerance in plants [30,31]. The use of the *Bradyrhizobium japonicum* strain has been reported to activate JA signaling, priming *Arabidopsis* for improved salt stress tolerance [32]. A *B. cereus* strain that is capable of producing VOCs and serine proteases has been found to confer tolerance to multiple abiotic stresses in plants. This tolerance is achieved through the modulation of the auxin, ABA, and JA signaling pathways, as well as the stimulation of antioxidant enzymes and secondary metabolites that have ROS scavenging capabilities [33].

Studies have shown that root exudates secreted from different plant species are able to attract various microorganisms to colonize their rhizosphere [34]. Furthermore, rhizobacteria from different environments have different abilities to produce bioactive metabolites that affect plant growth [35]. In this study, two *B. subtilis* strains isolated from different sources displayed distinct transcription profiles for genes related to the synthesis of antibiotic compounds, the siderophore metabolite bacillibactin, and the biofilm constituent EPS. These two bacterial strains showed similar effects in promoting root and shoot growth but differed in their effectiveness to induce plant tolerance against drought, heat, and copper stress. Both strains activated genes associated with the SA and JA signaling pathways and abiotic and biotic stress responses, as well as root growth and development. However, the expression of genes implicated in the ABA signaling pathway responded differently to treatment with BsCP1 and BsPG1.

## 2. Results

### 2.1. Molecular Identification of BsCP1 and BsPG1

The DNA sequences of the 16S ribosomal DNA fragments obtained from BsCP1 and BsPG1 were analyzed, revealing a phylogenetic relationship with different *Bacillus subtilis* strains (Figure 1). For further verification of these two bacterial strains, PCR analysis was performed to detect gene fragments involved in the synthesis of bacillibactin, including *DbhF*, *DbhB*, and *DbhE* [36], as well as *sfp*, which is responsible for the synthesis of surfactin [37], in the genomes of BsCP1 and BsPG1 (Figure 1B). The DNA sequences of these PCR fragments were subsequently analyzed using the BLAST program, showing 99% to 100% identity to sequences from different *B. subtilis* strains (Figure 1C).

### 2.2. Transcriptome Analysis of BsCP1 and BsPG1

BsCP1 and BsPG1 are two *B. subtilis* strains isolated from different environments. To predict bioactive metabolites produced by these two bacterial strains, RNA-seq analyses were employed to investigate their transcriptome profiles during both the log and stationary phases of culture. A total of 4458 genes from the BsCP1 culture and 4308 genes from the BsPG1 culture were analyzed. Of these, 60.8% and 76.9% of genes were up-regulated more than two-fold during the stationary phase in comparison to their expression levels in the log phase in the BsCP1 and BsPG1 cultures, respectively. For further analysis, gene transcripts involved in the synthesis of bioactive compounds linked to plant growth-promoting traits were selected. These genes participated in the production of seven types of antibiotics, biofilm components (e.g., EPS and spermidine), VOCs like terpenoids and 2,3-BD, the phytohormone IAA, the siderophore bacillibactin, serine proteases, and phosphate-solubilizing (PS) phosphatases. As a result, a total of 54 and 72 genes were identified in BsCP1 and BsPG1, respectively (Table 1). Their distribution within each gene group is shown in Figure 2A. Genes associated with the synthesis of various antibiotic compounds represented the largest group among genes correlated to plant growth-promoting traits in both bacterial strains, BsCP1 and BsPG1, accounting for 45% and 43%, respectively. Genes linked to the synthesis of IAA constituted the second largest group in both strains, representing 13%. Both bacterial strains contained genes responsible for producing seven antibiotic metabolites, including plipastatin (also known as fengycin), bacilysin, kanosamine, polyketide, phenazine, subtilosin, and surfactin (Figure 2B). Among these, in both BsCP1 and BsPG1, three major gene groups were found to be involved in the synthesis of plipastatin, subtilosin, and surfactin. However, the number of genes responsible for the synthesis of polyketide displayed a noticeable difference between BsCP1 and BsPG1: they constituted 13% in BsCP1 and 26% in BsPG1.

The genes identified from the transcriptome studies of BsCP1 and BsPG1 were classified as up-regulated when they had a Log_2_[FC] value greater than or equal to 1.0 and as not up-regulated when the Log_2_[FC] value was less than 1.0. The FC represents the fold change in expression levels during the stationary phase compared to the log phase. For these genes involved in synthesis of antibiotic compounds, the expression of genes responsible for synthesis of plipastatin, bacilysin, kanosamine, and phenazine exhibited similar patterns between BsCP1 and BsPG1 (Figure 3). However, there were distinct differences in gene expression patterns between the two bacterial strains regarding the synthesis of polyketide, subtilosin, and surfactin. In BsCP1, most genes involved in subtilosin synthesis were upregulated during the stationary phase. However, in BsPG1, subtilosin-synthesizing genes were not up-regulated. Distinct transcriptional patterns were observed for surfactin-synthesizing genes: while four genes in BsCP1 were not up-regulated, all six genes in BsPG1 were up-regulated. Both bacterial strains exhibited similar expression patterns for genes related to phosphate solubilizing activity, as well as the synthesis of IAA and terpenoids (Figure 3). For serine protease synthesizing genes, 50% and 100% of them were up-regulated in BsCP1 and BsPG1, respectively. In the context of bacillibactin synthesis during the stationary phase, BsCP1 had two genes: one with unchanged expression and another with reduced expression. In contrast, all seven genes in this category for BsPG1 were upregulated, each displaying a Log_2_ [FC] value greater than 1.0. For EPS synthesis, one out of four genes in BsCP1 showed increased expression. However, in BsPG1, eight genes exhibited upregulated expression. A single gene associated with 2,3-BD synthesis was upregulated in BsCP1 and not up-regulated in BsPG1. Similarly, one gene related to spermidine synthesis was identified in both strains; it was not up-regulated in BsCP1 but upregulated in BsPG1. The transcriptome analysis results reveal noticeable differences in transcription profiles between BsCP1 and BsPG1. This includes genes associated with the synthesis of antibiotics (e.g., polyketide, subtilosin, and surfactin), serine protease, bacillibactin, and EPS.

### 2.3. Biochemical Properties of BsCP1 and BsPG1

*Bacillus* species produce various metabolites that can act as biocontrol agents against phytopathogens [38]. After a seven-day coculture, BsCP1 and BsPG1 were able to suppress approximately 44% and 52% of the mycelial growth of *Foc TR4*, respectively (Figure 4A). *Foc* TR4 is a pathogen for causing banana *Fusarium* wilt [39]. However, when cultured on media separated from *Foc TR4*, the volatile metabolites produced by these two *B. subtilis* strains were observed to change the mycelia morphology, but had no effect on reducing the mycelial growth of *Foc TR4* (Figure 4B). Improving phosphate solubilization in the soil is a well-recognized mechanism for promoting plant growth mediated by PGPR [40]. BsPG1 displayed higher phosphate solubilizing activity than BsCP1 (Figure 4C). Both bacterial strains also produced between 0.8 to 5.6 ppm of IAA over a three-day culture period (Figure 4D). Despite this, both bacterial strains produced significant amounts of extracellular polymeric substances, EPS, with BsPG1 producing approximately 2.5 times more EPS compared to BsCP1 (Figure 4E). On the medium containing skim milk, similar clear zones were visible surrounding the colonies of BsCP1 and BsPG1, suggesting comparable levels of extracellular protease activity in both bacterial strains (Figure 4F). However, these strains exhibited different molecular patterns of extracellular protease in zymogram electrophoresis gel (Figure 4G). Based on the results of the transcriptome analysis and physiological characterization, BsPG1 was observed to have stronger transcriptional regulation for EPS synthesis genes. Correspondingly, BsPG1 accumulated a higher amount of EPS in its culture compared to BsCP1.

### 2.4. BsCP1 and BsPG1 Affected Growth of Arabidopsis Seedlings

Three-day-old *Arabidopsis* seedlings were co-cultured with BsCP1 and BsPG1 for 7 days. Both bacterial strains increased root branching and increased the number of lateral roots (Figure 5A). Consistently, the soil-grown *Arabidopsis* plants exhibited increased size and gained more fresh weight when treated with BsCP1 and BsPG1 (Figure 5B). Likewise, *Arabidopsis* seedlings treated with BsCP1 and BsPG1 showed increased lignin deposition (Figure 5C). Moreover, increased H_2_O_2_ accumulation was observed in the tissues treated with both BsCP1 and BsPG1, with BsPG1 inducing higher levels of H_2_O_2_ compared to BsCP1 (Figure 5D). However, the APX activity in BsCP1- and BsPG1-treated tissues was increased by 2.1-fold and 6.1-fold, respectively, compared to the control (Figure 5E). In contrast, the POD activity in BsCP1- and BsPG1-treated tissues was increased by 2.8-fold and decreased by 0.44-fold, respectively, compared to the control (Figure 5F). These results indicate that BsCP1 exhibited similar effects on activating the enzyme activities of APX and POD. However, BsPG1 had a strong effect on stimulating APX activity while reducing the activity of POD.

### 2.5. Differential Effects of BsCP1 and BsPG1 on Enhancing Plant Stress Resilience

Three-week-old *Arabidopsis* seedlings were pretreated with BsCP1 and BsPG1 inoculants and examined for drought stress tolerance by withholding water supply for 7 days. The seedlings that received BsCP1 and BsPG1 pretreatment exhibited a lower number of wilted seedlings, demonstrating a higher percentage of surviving plants (Figure 6A). In the post-drought period, the seedlings with BsCP1 and BsPG1 pretreatment displayed larger plant sizes and gained more fresh weight compared to the control seedlings (Figure 6B). The BsPG1-pretreated seedlings exhibited an elevated starch accumulation, whereas the BsCP1-treated seedlings did not show such an increase (Figure 6C). Starch serves as a storage form of photosynthate, and it is released under various abiotic stresses to act as an energy source and can alleviate stress damage [41]. Drought resistance is correlated with a higher accumulation of starch in common bean cultivars [42].

To analyze heat stress tolerance, two-week-old *Arabidopsis* seedlings were pretreated with BsCP1 and BsPG1 inoculants and subsequently exposed to a temperature of 45 °C for 20 min. Twenty-four hours after returning to the normal growth temperature of 23 °C, seedlings with the BsPG1 pretreatment showed a reduced number of wilted seedlings, indicating a higher percentage of survival compared to the control seedlings. However, the survival rate of seedlings pretreated with BsCP1 was similar to that of the control seedlings (Figure 7A). In the post-heat stress period, seven days after being cultured at 23 °C, seedlings with pretreatments exhibited larger plant sizes and gained more fresh weight. Specifically, the seedlings pretreated with BsPG1 showed a greater increase in fresh weight compared to the BsCP1-pretreated seedlings (Figure 7B). Furthermore, there was a significant increase in starch accumulation observed in both the BsCP1 and BsPG1 pretreated seedlings compared to the control seedlings. Importantly, the increment in starch accumulation was higher in the seedlings pretreated with BsPG1 compared to those pretreated with BsCP1 (Figure 7C).

Copper is one of the major heavy metal pollutants found in soil. Although a trace amount of copper is necessary for plant growth, an excessive quantity of copper induces oxidative stress, subsequently resulting in toxicity to plant biomass and chlorophyll content [43]. The response of seedlings to 200 µM CuSO_4_ was analyzed to determine the effects of BsCP1 and BsPG1 on improving heavy metal tolerance. As shown in Figure 8A, when exposed to 200 µM CuSO_4_, the control seedlings displayed symptoms of heavy metal damage in plants, such as leaf chlorosis and growth inhibition [44]. The control seedlings exhibited a 15.9% reduction in chlorophyll content and a 44.6% reduction in plant fresh weight under the 200 µM CuSO_4_ treatment (Figure 8B,C). The seedlings pretreated with BsCP1 showed a moderate enhancement in heavy metal tolerance, with a decrease in chlorophyll content by 7.6% and plant fresh weight by 24.4%. However, the seedlings pretreated with BsPG1 exhibited a stronger increase in copper stress tolerance, displaying only a minor reduction in chlorophyll content (0.7%) and plant fresh weight (4.9%).

### 2.6. BsCP1 and BsPG1 Altered Expression of Genes Associated with Hormone Signals

The treatments of BsCP1 and BsPG1 exhibited differential effects on improving the stress response of *Arabidopsis* seedlings. To investigate the underlying mechanisms, we examined the expression of genes associated with the SA, JA, and ABA signaling pathways using qPCR analysis. The functions of *isochorismate synthase 1* (*ICS1*), *enhanced disease susceptibility 1* (*EDS1*), and *CAM-binding protein 60-like G (CBP60G)* are involved in the synthesis and perception of SA [45,46,47]. The expression of all three genes was induced more than two-fold by BsCP1 and BsPG1 (Figure 9A). qPCR examined the expression of three genes involved in the JA synthesis and signaling pathway, including *oxophytodienoate-reducatase 3* (*OPR3*), *lipoxygenase 1* (*LOX1*), and *MYC2* [48,49,50]. The results show that treatment with BsCP1 resulted in a 2.1-fold increase in *OPR3* expression, a 2.4-fold increase in *LOX1* expression, and a 1.8-fold increase in *MYC2* expression. Conversely, treatment with BsPG1 led to a more substantial induction of gene expression associated with the JA signaling pathway, with all three genes showing an increase of more than two-fold in expression following BsPG1 exposure (Figure 9B). qPCR was also used to examine the expression of genes implicated in ABA synthesis and signaling transduction including *short-chain dehydrogenase reductase 4* (*SDR4*), *nine-cis-epoxycarotenoid dioxygenase 3* (*NCED3*), and *ABA insensitive 5* (*ABI5*) [51,52,53]. The results demonstrate that BsPG1 treatment led to a 2.2-fold, 2.1-fold, and 2.9-fold increase in *SDR4*, *NCED3*, and *ABI5* expression, respectively. In contrast, following BsCP1 treatment, the expression levels of *SDR4*, *NCED3*, and *ABI5* showed respective fold changes of 1.3, 0.36, and 0.87 compared to the control (Figure 9C). This result suggests that BsPG1 serves as a positive regulator of the ABA signaling pathway in *Arabidopsis*. However, this regulatory role was not evident for BsCP1.

Treatment using BsCP1 and BsPG1 enhanced tolerance toward various abiotic stress. The expression of genes involved in the regulatory network of abiotic stress responses such as drought and heat and oxidative stress induces the expression of *DREB2A* [54,55]. Likewise, *ZAT12* performs a role in responding to various abiotic stresses and oxidative stress [56]. Treatment with BsCP1 and BsPG1 increased the expression of both transcription factors (Figure 9D). BsCP1 treatment resulted in a 3.2-fold increase in *DREB2A* expression and a 10.2-fold increase in *ZAT12* expression. Similarly, BsPG1 treatment led to an 8.3-fold increase in *DREB2A* expression and a 2.3-fold increase in *ZAT12* expression. *Ascorbate peroxidase 1* (*APX1*) encodes a cytosolic form of APX playing a protective role for chloroplast under oxidative stress [57]. *Glutathione peroxidase 7* (*GPX7*) is a chloroplast form of GPX [58]. BsCP1 increased the expression of *APX1* by 2.7-fold and *GPX7* by 7.5-fold. BsPG1 increased *APX1* expression by 6.7-fold, but suppressed *GPX7* expression, showing a relative fold change of less than 1.0 (Figure 9D). *Delta-pyrroline-5-carboxylate synthase 1* (*P5CS1*) is involved in the synthesis of proline, acting as a osmoprotectant under abiotic stress [59]. *Chlorophyll a/b binding protein 1* (*CAB1*) functions in the light harvesting of photosystem II. This protein affects plant photosynthesis and fitness [60]. BsCP1 induced expression of CAB1 by 2.2-fold but failed to increase the amount of *P5CS1* transcription. BsPG1 showed significant induction for the expression of *P5CS1* and *CAB1 *(Figure 9D).

*flg22-induced receptor-like kinase 1* (*FRK1*) is a marker gene for elicitor response [61]. This gene is implicated in disease resistance that is induced by the presence of microbe-associated molecular patterns (MAMPs) [62]. SA and JA are two important plant metabolites to induce the expression of *pathogenesis-related* (*PR*) *proteins*, of which *PR-1* and *PR-2* are considered as the marker genes of the SA-induced signaling pathway [63]. However, the expression of *PR-3* and *thionin 2.1* (*Thi2.1*; *PR-13*) is controlled by the JA signal [64,65]. Both BsCP1 and BsPG1 induced expression of *FRK1*, *PR-1*, *PR-2* and *Thi2.1* more than two-fold (Figure 9E).

*Nitrilase 2* (*NIT2*) participates in the conversion of indole-3-acetonitrile into the phytohormone IAA, a crucial regulator in the development of lateral roots [2]. The enzyme activity of the NADPH oxidase/respiratory burst oxidase homolog (RBOH) is involved in localized production of oxidative bursts, which regulate plant development and stress responses [66]. ROS produced by RBOHC play roles in regulating root response in developmental stages and environmental conditions and in controlling the growth of primary roots and root hairs [67,68]. Both BsCP1 and BsPG1 increased transcripts of *NIT2* and *RBOHC* more than two-fold. Moreover, BsPG1 exerted a strong effect on gene induction of *RBOHC* (Figure 9F).

## 3. Discussion

Plant roots secrete various metabolites that act as nutrient sources and environmental signals for rhizosphere bacteria [69]. The composition of these root exudates depends on factors such as the genetic makeup of the plant species, its developmental stage, nutrient availability, and prevailing environmental conditions. Consequently, these exudates can significantly shape the microbial community within the rhizosphere [34]. This study examined the transcription profiles of genes related to plant growth-promoting traits in two *B. subtilis* strains isolated from different sources. *B. subtilis* is renowned for its ability to produce a wide array of metabolites with antimicrobial properties [70]. Our findings reveal that both bacterial strains expressed a substantial number of genes that are linked to the synthesis of antibiotic compounds including plipastatin, bacilysin, kanosamine, polyketide, phenazine, subtilosin, and surfactin. However, these two bacterial strains exhibited differential transcriptional regulation for certain antibiotic synthesis genes, including those involved in the synthesis of polyketide, subtilosin, and surfactin. Consistently with this, a previous study showed that *B. subtilis* strains sourced from different ecological sites may have distinct antibiotic profiles [35]. Our transcriptome analysis identified genes tied to the production of seven antimicrobial metabolites in BsCP1 and BsPG1. Among these, the antifungal activities of plipastatin and surfactin have been validated in previous studies [35,71]. Our results show that BsCP1 and BsPG1 exhibit similar levels of antifungal activity against the pathogen of banana *Fusarium* wilt. The genes responsible for plipastatin synthesis were dominant in both BsCP1 and BsPG1. However, BsPG1 displayed positive transcriptional regulation for surfactin-synthesizing genes, while BsCP1 showed negative transcriptional regulation for these gene expressions. A study by Kiesewalter et al. [35] highlighted the effectiveness of plipastatin in inhibiting the growth of *Fusarium* spp. mycelia. Thus, plipastatin is likely the primary factor responsible for the antifungal efficacy against the *Foc* TR4 pathogen observed in both strains.

The transcriptome data revealed similar transcription profiles for both IAA-synthesizing genes and the genes encoding enzymes responsible for phosphate solubilization in BsCP1 and BsPG1. Consistently with this, both bacterial strains were confirmed to produce IAA and exhibit phosphate solubilizing activity. The application of both bacterial strains to *Arabidopsis* seedlings resulted in increased root branching and stimulated seedling growth. Moreover, at the transcription level, BsCP- and BsPG1 treatment increased the expression of *NIT2*, which is involved in IAA synthesis, as well as *RBOHC*, which plays a role in root hair development by modulating the ROS signal [72,73]. IAA production and phosphate solubilizing activity represent two important traits in PGPR bacterial strains that promote plant growth [74]. Thus, the activities related to IAA production and phosphate solubilization may play a part in the plant growth-promoting effect mediated by BsCP1 and BsPG1. Furthermore, BsCP1 and BsPG1 are capable of generating bioactive VOCs. This is evidenced by the transcriptional control of genes linked to the production of terpenoids and 2,3-BD, as well as their ability to produce volatile substances that affect fungal growth. Although showing polymorphic zymogram patterns for extracellular proteases, these two bacterial strains showed similar extracellular protease activity on the milk-containing medium. At the transcriptional level, treatments with both bacterial strains elevated the expression of genes associated with the SA and JA signaling pathways, as well as genes related to induced disease resistance, such as *FRK1* and *PR* genes. Microbial VOCs have been shown to enhance plant tolerance against both abiotic and biotic stress by activating the SA and JA signaling pathways [75]. Ling et al. [76] demonstrated the antimicrobial activity of microbial serine proteases. Additionally, serine proteases derived from rhizobacteria have been observed to activate plant immune responses and modify cell wall lignification [77]. Studies have shown the significance of SA and JA signaling in mitigating plant oxidative stress, thereby enhancing stress tolerance [24,27]. Therefore, metabolites and enzymes like VOCs and serine proteases from BsCP1 and BsPG1 could play a pivotal role in triggering the SA and JA signaling pathways. These activations can bolster antioxidant defenses and fortify plants against challenges such as drought, heat, and copper stress.

BsCP1 and BsPG1 showed different transcription regulation for genes responsible for the synthesis of bacillibactin and EPS. Bacillibactin is an iron-binding metabolite from the *Bacillus* species. Its role in plant growth is emphasized by facilitating nutrient uptake efficiency and controlling plant disease through its antimicrobial activity and the activation of systemic resistance [78,79]. During the stationary phase, the majority of genes associated with the synthesis of EPS were up-regulated in BsPG1, but not in BsCP1. The discrepancy in gene expression associated with the synthesis of EPS was consistent with the results showing that BsPG1 produced more EPS than BsCP1. EPS is a main component of biofilm and the production of EPS in microorganisms is affected by environmental factors [80]. Research has reported that the production of EPS is necessary for establishing successful symbiosis between nitrogen-fixing bacteria and host plant roots [14,81]. Since BsPG1 was derived from the rhizosphere of peanut plants, it may account for its increased capacity to generate more EPS than BsCP1.

EPS-producing PGPR may enhance plant tolerance to abiotic stress through various mechanisms, such as retaining soil moisture, triggering antioxidant activity, and the accumulation of osmolytes, as well as removing toxic heavy metals [82,83,84]. Compared to BsCP1, BsPG1 was a stronger inducer for APX activity and gene expression of *APX1*. The critical function of *APX1* within the regulatory network linked to oxidative stress has been established [57]. In this study, treatment with BsPG1 conferred greater tolerance to drought, heat, and copper stress than treatment with BsCP1 did. Increased oxidative damage is a common outcome resulting from drought, heat, and copper stress [16,17,18]. Given its higher EPS production, BsPG1 might activate APX1 activity more effectively than BsCP1, which could be a crucial factor in alleviating the harmful impacts of various abiotic stresses. Moreover, BsPG1 exhibited a higher efficiency in inducing starch accumulation during the post-drought and post-heat stress periods. Seedlings treated with BsPG1 also displayed reduced chloroplast damage under copper stress. Starch serves as a storage form of the photosynthetic product. Remobilizing starch to release energy can assist plants in alleviating stress damage [85]. Sugars released from starch under abiotic stress conditions function not only as energy sources but also act as osmoprotectants against osmotic stress [86]. Elevated levels of starch have been detected in *Arabidopsis* leaf tissues when exposed to temperature and osmotic stress conditions [87,88,89]. The ABA signal is a positive regulator of starch metabolism in maize and rice [90,91]. In *Arabidopsis*, the expression of ADP-glucose pyrophosphorylase (AGPase), a key enzyme involved in starch synthesis, is stimulated by sucrose [92]. Furthermore, ABA has the ability to enhance starch synthesis driven by sucrose [93]. Exogenous ABA has been reported to induce the expression of genes involved in starch synthesis, leading to increased starch accumulation in grapevine cuttings [94]. Consistently, our qPCR results reveal that BsPG1 could activate *Arabidopsis* genes involved in ABA synthesis, such as *NCED3* and *SDR4*, as well as those related to ABA perception like *ABI5*. However, the expression of these genes was not up-regulated in seedlings treated with BsCP1. *P5CS1* is involved in the synthesis of the osmolyte proline [95]. The *P5CS* transcript is induced by drought and salinity stress and the presence of ABA. Additionally, *P5CS* gene activation in seedlings under salt stress is negated in the ABA-deficient mutant [96]. In line with this, only BsPG1 up-regulated the expression of *P5CS1*, while BsCP1 had no effect on its expression. Thus, BsPG1 might significantly enhance abiotic tolerance by modulating ABA signaling in *Arabidopsis* seedlings. The superior EPS production by BsPG1 could stimulate the ABA signaling pathway, which could enhance resistance to stress by minimizing chlorophyll degradation and augmenting the accumulation of starch and proline when subjected to drought, heat, and copper stress.

## 4. Materials and Methods

### 4.1. Bacterial Strain Isolation and Characterization

The soil sample from commercial compost and peanut roots were dissolved in sterilized water. The obtained supernatants were then spotted on the nutrient agar (NA) medium and incubated at 30 °C for overnight. PCR amplification of the genomic DNA from bacterial strains isolated from compost or nodule supernatant was performed using the primers fD1 and rP1 to detect the 16S rDNA sequence. The primers DbhB, DbhE, and DbhF were used for genes associated with bacillibactin synthesis, and sfp was used for surfactin synthesis. PCR amplification began with an initial denaturation at 95 °C for 5 min, followed by 30 cycles of 95 °C for 1 min, annealing at 52 °C for 30 s, and extension at 72 °C for 5 min. This was concluded with a final extension at 72 °C for 3 min. The sequencing results of amplified PCR fragments were analyzed using the Basic Local Alignment Search Tool (BLAST) program [97]. The phylogenetic tree of closely related bacterial strains was constructed using the neighbor-joining algorithm (NJ) in MEGA X software [98]. Primer sequences used for PCR analysis are listed in Appendix A.

### 4.2. RNA-Seq Analysis

Bacteria were cultured for 4 and 16 h, representing the log and stationary phases, respectively. The cells were collected for RNA extraction using the hot SDS/phenol method as described by Jahn et al. [99]. We subjected 1 µg of total RNA to library construction using the Universal Prokaryotic RNA -Seq library preparation kit (TECAN). Sequencing was performed using a NovaSeq 6000 System (Illumina, San Diego, CA, USA). The quality of RNA-seq raw reads was evaluated using CLC Genomics Workbench 10 software (Qiagen, Germantown, MD, USA). The raw reads were trimmed and assembled using SPAdes (v3.15.3) [100]. The rRNA, tRNA, and open reading frame (ORF) of the protein coding sequence were predicted using RNAmmer (v1.2), tRNAscan-SE (v1.3.1), and the Glimmer program, respectively [101,102,103]. The ORFs were annotated using NCBI blast software (v.2.2.28+) and the COGs (Clusters of Orthologous Groups) database [104]. The gene ontology (GO) annotation was performed using FastAnnotation, while pathway analysis was conducted through the KEGG Automatic Annotation Server (KAAS) [105,106]. The prediction of pathways and antibiotic gene prediction were carried out using the CARD database [107]. The gene expression values were quantified using FPKM (Fragments Per Kilobases per Million). The differential gene expression between the two time points was determined by dividing the FPKM value of the 16 h culture by that of the 4 h culture.

### 4.3. Analysis of Biochemical Properties of Isolated Bacterial Strains

#### 4.3.1. Antifungal Activity

Detecting the antifungal activity of the nonvolatile and volatile metabolites produced by bacterial strains was performed using methods described by Tsai et al. [33]. In brief, to detect nonvolatile compounds, two filter paper pieces were prepared, each containing 10 µL of bacterial solutions that had been cultured overnight in LB medium at a concentration of 1 × 10^8^ CFU/mL, while water was used as a control. These filter papers were placed 3 cm away from a mycelial plug of *Fusarium oxysporum* f. sp. *cubense* tropical race 4 (*Foc* TR4) on potato dextrose agar (PDA) medium, and the cultures were cocultured for 7 days at 28 °C. To test the antifungal activity of volatile metabolites produced by the bacterial strains, both the fungal pathogen *Foc* TR4 and the bacterial strains were separately grown on Petri dishes containing PDA medium and LB medium, respectively. Subsequently, the plate containing the fungal strain and bacterial strains were positioned face-to-face and cocultured for 7 days at 28 °C. The diameter of the fungal mycelia in the control group and those cocultured with bacterial strains was measured to determine the rate of inhibition of mycelial growth (I) = (1-diameter of mycelia of treatment/diameter of mycelia of control) × 100.

#### 4.3.2. Quantification of Phosphate Solubilization

To quantify the activity of phosphate solubilization, a single colony of the tested bacterial strain was grown in Pilovskaya’s (PVK) medium containing 1% glucose, 0.05% (NH_4_)_2_SO_4_, 0.02% NaCl, 0.03% KCl, 0.052% FeSO_4_.7H_2_O, 0.045% MnSO_4_.4H_2_O, 0.057% MgSO_4_.7H_2_O, 0.5% Ca_3_(PO_4_)_2_, and 0.04% yeast extract at 28 °C for 3 days. The bacterial culture was then subjected to centrifugation, and 1 mL of the resulting supernatant was combined with 0.4 mL of Vanadate–Molybdate reagent and 0.6 mL of H_2_O. The mixture was then incubated at room temperature for 60 min, and the optical density at 470 nm was recorded. The concentration of solubilized phosphate in the bacterial supernatant was calculated based on a standard curve prepared from various concentrations of KH_2_PO_4_.

#### 4.3.3. Detection of Indole Acetic Acid (IAA) and Exopolysaccharide (EPS) Production

For the quantification of indole acetic acid (IAA), the tested bacterial strains were cultured in Luria–Bertani (LB) medium containing 2 mM tryptophan for 24, 48, and 72 h at 28 °C. One milliliter of bacterial supernatant was mixed with the Salkowski reagent and incubated at room temperature for 10 min. The absorbance of the mixture was measured at a wavelength of 530 nm to determine the concentration of IAA.

#### 4.3.4. Detection of Exopolysaccharide (EPS) Production

The EPS production of the tested bacterial strains was assessed using the methods described by Nwosu et al. [108]. The bacterial strains were cultured in nutrient broth containing 0.5% peptone, 0.3% yeast extract, and 0.5% NaCl, supplemented with 2% sucrose. After incubating the cultures at 28 °C for 24 h, they were subjected to centrifugation. The resulting supernatants were combined with two volumes of 95% ethanol and incubated overnight at 4 °C. Subsequently, the EPS extracted from the bacterial supernatants was collected via centrifugation and dried for 3 h at 60 °C. The amount of EPS produced by the bacterial strains was determined by measuring the dry weight of the EPS precipitates.

#### 4.3.5. Analysis of Protease Activity

To detect protease activity, a 5 µL aliquot of a bacterial culture with a concentration of 1 × 10^8^ CFU/mL was placed on a 5 mm diameter filter paper positioned at the center of a medium containing 5% skim milk and 4% agar. The culture was then incubated at 28 °C for 2 days. The presence of a clear zone around the bacterial colony indicates the presence of protease activity. For the preparation of zymogram analysis, bacterial strains were cultured in a medium containing 1% peptone, 0.5% yeast extract, and 1% skim milk at 28 °C for 3 days. Subsequently, the bacterial cultures were centrifuged to collect the cell-free supernatants. The protease pellets were precipitated by adding ammonium sulfate until reaching 80% saturation. Afterward, the protease pellets were collected through centrifugation, suspended in 50 mM of phosphate buffer at pH 7.0, and subsequently used for zymogram electrophoresis analysis, following the method described by Tsai et al. [33].

### 4.4. Analysis of Growth-Promoting Effects in Arabidopsis Seedlings

Four-day-old seedlings of *Arabidopsis thaliana* (Columbia ecotype) obtained from The Arabidopsis Information Resource (TAIR) were co-cultured with the tested bacterial colonies. These colonies were positioned 4 cm away from the seedlings in 1/2 Murashige and Skoog (MS) medium and then incubated at 23 °C under a 16 h lighting condition. The number of lateral roots in both the control and treated seedlings was recorded after a seven-day co-culture period. Further growth-promoting effects of the tested bacterial strain were observed in *Arabidopsis* seedlings grown in soil. The tested bacterial strain cultured in the LB broth for 16 h was collected via centrifugation and suspended in water to a bacterial density of 1 × 10^8^ CFU/mL. The bacterial solution was administered through foliar spraying on 2-week-old *Arabidopsis* seedlings once a week for three successive weeks. In the control, seedlings were treated with water. Twenty seedlings were included in each treatment. The fresh weights of both the control and treated seedlings were recorded, and lignin was extracted from the tissues of both groups following the methods described by Govender et al. [109]. Briefly, leaf tissues of 0.1 g were extracted in a solution containing 100 mM phosphate buffer (pH 7.4) and 0.5% Triton-100. The obtained pellets were washed with 95% methanol and dried at room temperature. The crude lignin extracts were used for lignin quantification using the thioglycolic acid (TGA) method as described by Bruce and West [110]. To detect H_2_O_2_ production, 0.1 g of leaf tissues were extracted in 80% ethanol, and the obtained supernatants were mixed with the ferrous ion oxidation xylenol orange (FOX) reagent as described by DeLong et al. [111]. After incubation for 30 min in the dark, the H_2_O_2_ concentrations were determined by measuring the absorbance at 560 nm. The enzyme activity of ascorbate peroxidase (APX) was quantified by extracting 0.1 g of leaf tissues in a solution containing 50 mM potassium phosphate buffer (pH 7.0), 1% polyvinylpyrrolidone (PVP), and 0.1 mM EDTA, as described by Nakano and Asada [112]. The obtained supernatants were mixed with a reaction solution containing 50 mM potassium phosphate buffer (pH 7.0), 0.2 mM EDTA, 0.5 mM ascorbic acid, and 2% H_2_O_2_. APX activity was determined by measuring the absorbance at 290 nm after 5 min of incubation. To detect guaiacol peroxidase (POD) activity, 0.1 g of leaf tissues were extracted in a solution containing 0.1 mM EDTA and 0.2 M potassium phosphate buffer (pH 7.8), as described by Aebi [113]. The obtained supernatants were mixed with a reaction solution containing 1% guaiacol, 40 mM H_2_O_2_, and 100 mM phosphate buffer (pH 7.0). POD activity was quantified by measuring the absorbance at 470 nm after incubation for 10 min. The experiments were conducted three times.

### 4.5. Analysis of Abiotic Stress Tolerance in Arabidopsis Seedlings

#### 4.5.1. Analysis of Drought Stress Tolerance

For the analysis of drought stress tolerance, three-week-old *Arabidopsis* seedlings grown in seedling trays were pretreated with a bacterial suspension of 1 × 10^8^ CFU/mL 3 times. After bacterial treatments, watering was withheld from the seedlings for 7 days. The watering was resumed for the drought-stressed seedlings for 5 days. Forty seedlings were included in each treatment. The fresh weight and starch contents were analyzed at the end of 5-day culture. The survival rates of the control and treated seedlings were calculated by dividing seedlings without wilted leaves by total seedling number. Seedling fresh weights and starch contents were measured at the end of the 5-day recovery period. The starch contents were analyzed using a method described by Tsai et al. [114]. Briefly, seedlings were boiled in water for 5 min and then transferred to 80% ethanol for an additional 3-min boiling. For a qualitative assay, 100 μL of Lugol’s iodine solution was used to stain starch granules in the leaf tissues. To quantify starch content, the solution was mixed with Lugol’s iodine solution (Sigma-Aldrich, St. Louis, MI, USA) and the absorbance at 620 nm was measured. The drought stress tolerance analysis was performed in triplicate.

#### 4.5.2. Analysis of Heat Stress Tolerance

To analyze the heat stress response, two-week-old *Arabidopsis* seedlings treated with the tested bacterial strains were exposed to 45 °C for 20 min and then returned to a growth temperature of 23 °C. Each treatment consisted of 40 seedlings. The wilted seedlings were observed after recovery at 23 °C for 24 h to calculate their survival rates. Seedling fresh weights and starch contents were measured at the end of the 7-day recovery period. The starch contents were analyzed using a method described above. The heat stress tolerance analysis was performed in triplicate.

#### 4.5.3. Analysis of Copper Stress Tolerance

For testing plant tolerance to copper stress, two-week-old *Arabidopsis* seedlings were pretreated with bacterial suspension in 1 × 10^8^ CFU/mL 3 times, while the control plants were treated with water. One day after the pretreatment, a 20 mL solution of 200 µM CuSO_4_ was applied to the soil once every two days for a total of three times. Each treatment consisted of 30 seedlings. One day after the final treatment, the fresh weight of both CuSO_4_-treated and untreated plants was measured. The growth inhibition rate (%): (1-fresh weight of CuSO_4_-treated plants/fresh weight of water-treated plants) × 100. The chlorophyll contents of both CuSO_4_-treated and untreated plants were extracted and quantified using a method described by Kurniawan and Chuang [115]. The relative chlorophyll reduction rate (%): (1-chlorophyll content of the CuSO_4_-treated plants/ chlorophyll content of water-treated plants) × 100. The analysis for copper stress tolerance was conducted three times.

### 4.6. qPCR Analysis for Arabidopsis Gene Expression

Total RNA was extracted from the leaf and root tissues of 10-day-old *Arabidopsis* seedlings cocultured with the tested bacterial strains for 7 days using the methods described by Lee Downing et al. [116]. cDNA was prepared from 2 μg of total RNA using ImProm-II™ reverse transcriptase (Promega) for qPCR amplification. The amplification was performed using SYBR Green Master Mix in the StepOneTM Real-Time PCR System (Thermo Fisher). The relative gene expression level was calculated using the 2^−ΔΔCt^ method and the gene expression of *actin 2* (*ACT2*) was used as a reference gene for normalization. The primer sequences used for qPCR analysis are listed in Appendix A.

### 4.7. Statistical Analysis

The differences between the treatments were analyzed using the ANOVA procedure in the SAS (3.8) software package. A *p*-value less than 0.05 from the Tukey test was considered to be statistically significant.

## 5. Conclusions

This research investigated two *B. subtilis* bacterial strains sourced from distinct origins. Both strains demonstrated similarities in antimicrobial activity, extracellular protease function, phosphate solubilization, and the production of IAA and VOCs. Yet, differences emerged in their synthesis of bacillibactin and EPS. Both bacterial strains enhanced lateral root development and increased seedling fresh weight; furthermore, they activated the antioxidant activity and plant defense response to increase tolerance against various abiotic stresses. Our results indicate that *B. subtilis* strains with a greater ability to produce EPS show a stronger capacity to enhance plant tolerance to drought, heat, and copper stress. Additionally, strains that generate larger quantities of EPS are capable of triggering the ABA signaling pathway in *Arabidopsis* seedlings. The results of this study suggest that PGPR strains from different ecological sites have significant potential to produce polymorphic metabolites, which can vary in their effects on plant growth and stress response.

## Figures and Tables

**Figure 1 ijms-24-13720-f001:**
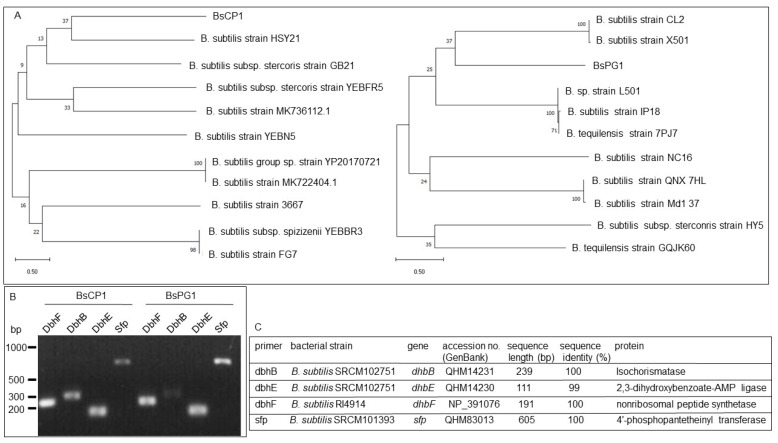
Characterization of BsCP1 and BsPG1. The 16S rDNA sequences from various *B. subtilis* strains were used for construction of a phylogenetic tree for BsCP1 (left) and BsPG1 (right) (**A**). Electrophoresis of PCR products generated from the genomic DNA of BsCP1 and BsPG1 (**B**). BLAST results for the gene segments derived from PCR amplification (**C**).

**Figure 2 ijms-24-13720-f002:**
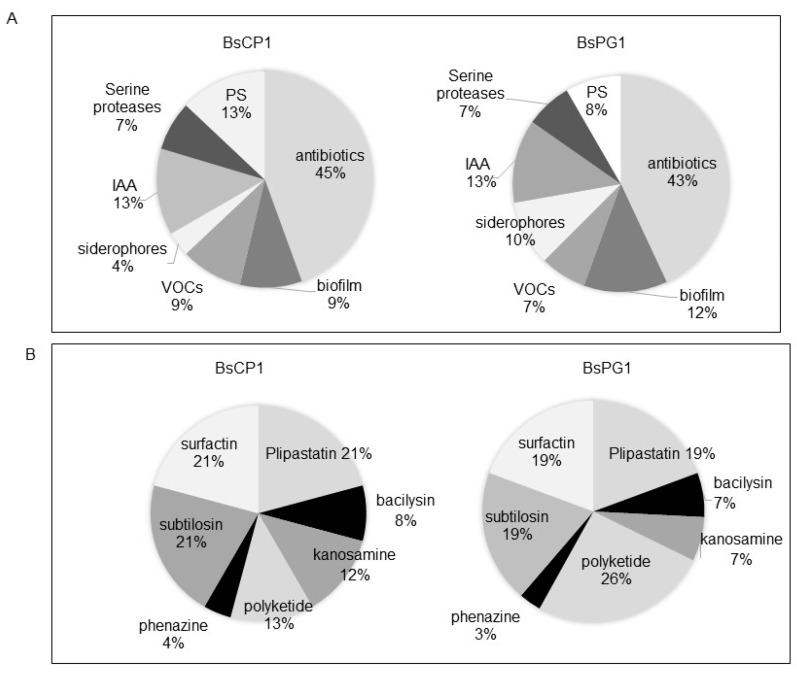
Genes associated with plant growth-promoting traits in BsCP1 and BsPG1 genome. These genes were classified into seven groups based on their annotated functions (**A**). Genes involved in antibiotic synthesis were further subgrouped into seven categories (**B**). PS: phosphate solubilization. VOCs: volatile organic compounds.

**Figure 3 ijms-24-13720-f003:**
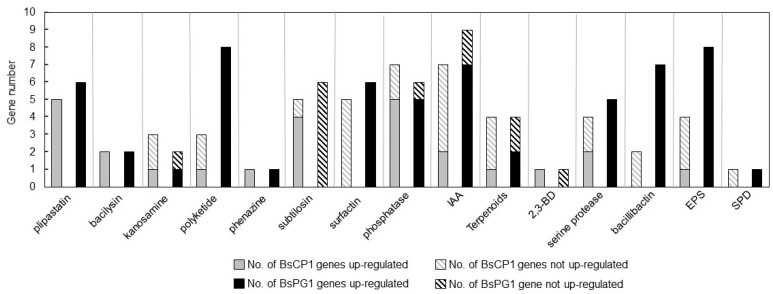
Comparison of genes derived from transcriptome analyses. Genes identified from the transcriptome analysis of BsCP1 and BsPG1, associated with the synthesis of metabolites that correlate with plant growth promotion, were categorized based on their fold change in expression levels during the stationary phase compared to the log phase. Genes with a Log_2_[FC] value of 1.0 or higher were considered up-regulated. These are represented by a gray solid block for BsCP1 genes and a black solid block for BsPG1 genes. Genes with Log_2_[FC] values less than 1.0 are represented by a gray striped block for BsCP1 genes and a black striped block for BsPG1 genes. These values signify genes that are not up-regulated in expression during the stationary phase compared to the log phase.

**Figure 4 ijms-24-13720-f004:**
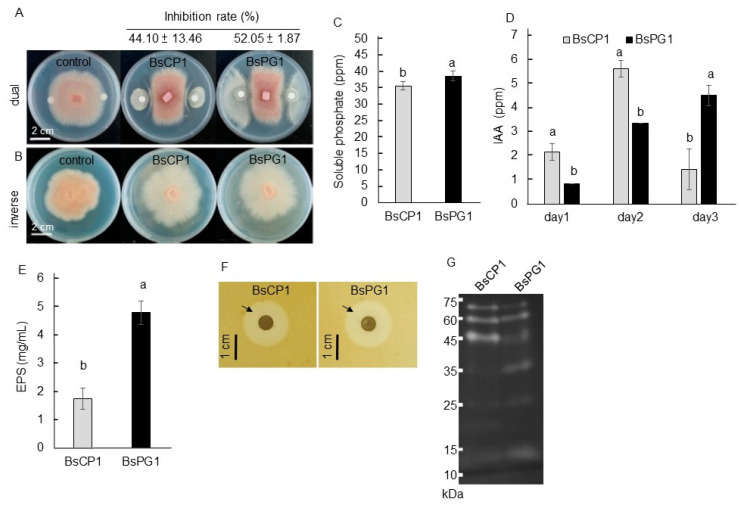
Physiological properties of strains BsCP1 and BsPG1. Both bacterial strains produced diffusible metabolites to suppress the mycelial growth of *Foc* TR4 (**A**). Volatile compounds produced by both bacterial strains altered the mycelial growth of *Foc* TR4 (**B**). Phosphate solubilizing activity was detected in the bacterial culture (**C**). IAA concentrations detected in bacterial culture (**D**). Exopolysaccharides (EPS) were produced by the bacterial strains (**E**). Clear zones (indicated by arrows) surrounding the bacterial colonies suggest protease activity (**F**). Zymogram gel was used to detect extracellular proteases on a native gel electrophoresis, using casein as a substrate (**G**). Values in each histogram represent the mean of three replicates ± SD. In histograms C, D, and E, different letters indicate statistical significance at *p* = 0.05.

**Figure 5 ijms-24-13720-f005:**
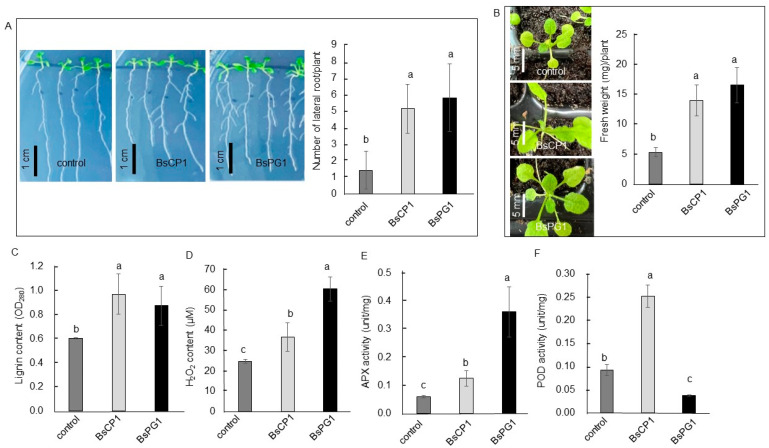
Physiological responses of *Arabidopsis* seedlings toward BsCP1 and BsPG1 treatment. Root architecture and lateral root number in seedlings cocultured with BsCP1 and BsPG1 inoculants for seven days (**A**). Soil-grown seedlings were treated with BsCP1 and BsPG1 once a week. After three weeks of treatment, the seedlings’ growth conditions and changes in fresh weight were analyzed (**B**), as well as their lignin content (**C**), H_2_O_2_ content (**D**), APX activity (**E**), and POD activity (**F**) in the leaves. Values in each histogram represent the mean of three replicates ± SD. Different letters within the histograms represent statistical significance at a *p*-value of 0.05.

**Figure 6 ijms-24-13720-f006:**
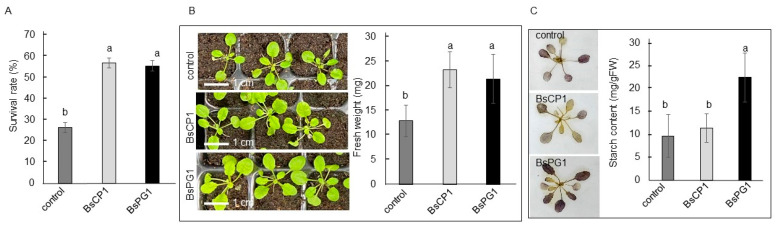
Drought stress tolerance induced by BsCP1 and BsPG1. Three-week-old seedlings treated with BsCP1 and BsPG1 had watering withheld for seven days. After resuming growth at 23 °C for five days, the drought-stressed seedlings were investigated for survival rate (**A**), seedling size and fresh weight (**B**), and starch accumulation by staining and quantitative measurement (**C**). Values in each histogram represent the mean of three replicates ± SD. Different letters within the histograms represent statistical significance at a *p*-value of 0.05.

**Figure 7 ijms-24-13720-f007:**
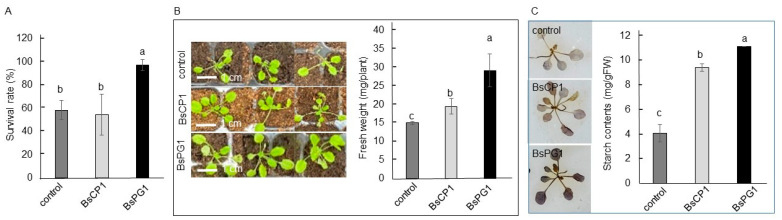
Heat stress tolerance induced by BsCP1 and BsPG1. Two-week-old *Arabidopsis* seedlings treated with BsCP1 and BsPG1 were exposed to 45 °C for 20 min. Twenty-four hours after returning to 23 °C growth condition, the survival rate of the heated-stressed seedlings was analyzed (**A**). Seven days after recovering at 23 °C, seedlings were investigated for seedling size and fresh weight (**B**), and starch accumulation by staining and quantitative measurement (**C**). Values in each histogram represent the mean of three replicates ± SD. Different letters within the histograms represent statistical significance at a *p*-value of 0.05.

**Figure 8 ijms-24-13720-f008:**
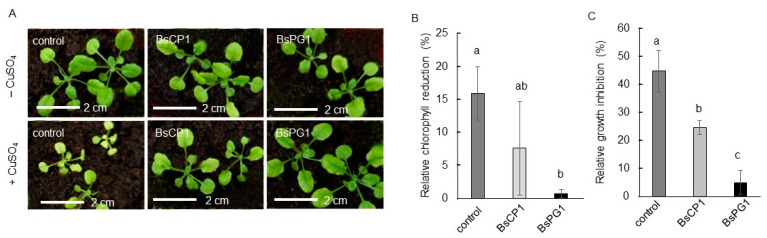
Analysis of copper stress tolerance induced by BsCP1 and BsPG1. Two-week-old *Arabidopsis* seedlings treated with BsCP1 and BsPG1 were exposed to 200 µM CuSO_4_. After three exposures, obvious stunting of growth and leaf bleaching were observed in the control seedlings (**A**). The relative chlorophyll reduction (**B**) and growth inhibition rate (**C**) were investigated. Values in each histogram represent the mean of three replicates ± SD. In histograms B and C, different letters represent statistical significance at a *p*-value of 0.05.

**Figure 9 ijms-24-13720-f009:**
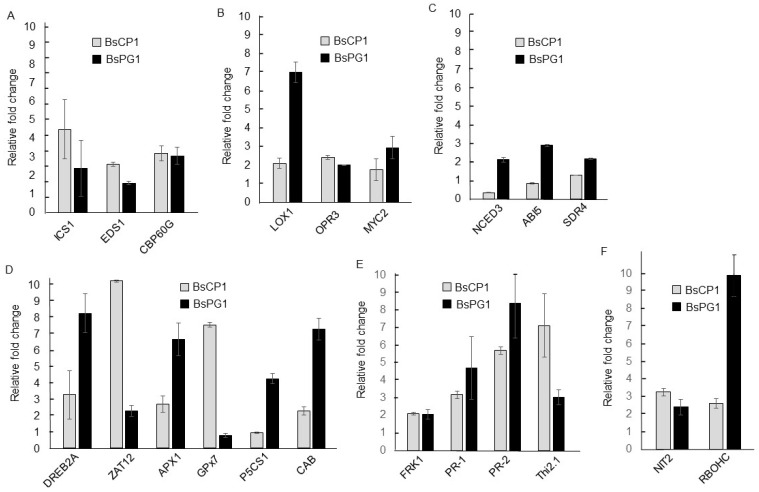
*Arabidopsis* gene expression altered by BsCP1 and BsPG1. Changes in gene expression were detected via qPCR analysis using total RNA prepared from whole seedlings treated with BsCP1 and BsPG1. Genes associated with various pathways were investigated, including (**A**) SA signaling pathway: *isochorismate synthase 1* (*ICS1*), *enhanced disease susceptibility 1* (*EDS1*), and *CAM-binding protein 60-like G* (*CBP60G*). (**B**) JA signaling pathway: *oxophytodienoate-reducatase 3* (*OPR3*), *lipoxygenase 1* (*LOX1*), and *MYC2*. (**C**) ABA signaling pathway: *short-chain dehydrogenase reductase 4* (*SDR4*), *nine-cis-epoxycarotenoid dioxygenase 3* (*NCED3*), and *ABA insensitive 5* (*ABI5*). (**D**) Abiotic stress response: *DREB2A*, *ZAT12*, *ascorbate peroxidase 1* (*APX1*), *glutathione peroxidase 7* (*GPX7*), *delta-pyrroline-5-carboxylate synthase 1* (*P5CS1*), and *chlorophyll a/b binding protein 1* (*CAB1*). (**E**) Disease resistance: *flg22-induced receptor-like kinase 1* (*FRK1*), *pathogenesis-related* (*PR*) *protein 1* (*PR-1*), *PR-2*, *PR-3*, and *thionin 2.1* (*Thi2.1*). (**F**) Root development: *nitrilase 2* (*NIT2*) and *NADPH oxidase/respiratory burst oxidase homolog C* (*RBOHC*).

**Table 1 ijms-24-13720-t001:** Genes related to plant growth-promoting traits.

Acc. No.	Gene	Log_2_[FC]	Acc. No.	Gene	Log_2_[FC]
BsCP1			BsPG1		
Plipastatin					
WP_089172562	*PpsC*	3.2	WP_086343904	*PpsC*	3.5
WP_032723105	*PpsB*	3.0	WP_129092450	*PpsB*	3.1
WP_160214989	*PpsA*	4.2	WP_101169517	*PpsA*	2.9
WP_129092448	*PpsD*	3.8	WP_080262617	*PpsE*	4.2
WP_186453377	*PpsD*	2.4	WP_128737913	*PpsD*	3.5
			WP_129092448	*PpsD*	4.5
Bacilysin					
WP_003244300	*BacB*	1.8	WP_003244300	*BacB*	2.2
WP_032722711	*BacD*	1.9	WP_124059367	*BacD*	1.2
Kanosamine					
WP_032721285	*NtdB*	0.5	WP_101169444	*NtdA*	0.7
AFQ56969	*NtdC*	7.9	WP_024572383	*NtdB*	0.5
WP_019712355	*NtdA*	0.7			
Polyketide					
WP_080287605	*PksL*	0.9	WP_185184354	*PksL*	3.5
WP_003231805	*PksG*	2.0	WP_024573082	*PksG*	3.2
WP_124048390	*PksF*	0.1	WP_173614094	*PksF*	3.0
			WP_124059875	*PksL*	2.9
			TDY57959	*PksN*	2.8
			AGZ20286	*PksD*	3.2
			WP_167559687	*PksJ*	3.1
			AGZ20287	*PksD*	3.0
Phenazine					
WP_032723009	*PhzF*	1.5	WP_069837383	*PhzF*	1.9
Subtilosin					
WP_019712818	*AlbD*	−0.1	WP_123374486	*AlbD*	0.7
WP_003222006	*AlbB*	2.3	QHF59890	*Syn. Pro*	0.8
WP_032722691	*AlbA*	1.3	WP_003222006	*AlbB*	0.4
WP_003222002	*BesA*	6.1	WP_123374484	*AlbA*	−1.6
WP_015250988	*AlbG*	1.2	WP_003222002	*Sub. A*	−3.0
			WP_021480840	*AlbG*	0.8
Surfactin					
WP_144481589	*SrfAA*,	−4.1	WP_137200567	*SrfAA*	2.3
WP_029726578	*SrfAD*	−2.6	WP_185184456	*SrfAD*	3.2
WP_032722905	*SrfAC*	−4.2	WP_185184457	*SrfAC*	1.4
WP_160215003	*SrfAA*	−2.7	WP_167559147	*SrfAA*	1.3
WP_015715234	*Sfp*	0.0	WP_003234549	*sfp*	3.1
			WP_129092244	*SrfAB*	1.0
Phosphatase					
WP_003245272	*PhoH*	2.6	WP_080009778	*PhoA*	5.4
WP_010886458	*PhoA*	2.1	WP_101169869	*PhoD*	6.7
WP_032722881	*PhoD*	0.8	WP_076458498	*PhoB*	5.6
WP_080287651	*PhoB*	1.6	WP_014476350	*PhoE*	3.1
WP_003233157	*PhoE*	1.0	WP_014477373	*PhoH*	0.7
WP_014477373	*PhoH*	0.6	WP_129092478	*phytase*	6.2
WP_003230820	*phytase*	6.6			
IAA					
WP_032722039	*TrpC*	0.5	WP_003230601	*TrpC*	3.1
WP_003245959	*TrpD*	0.7	WP_134981823	*TrpD*	3.7
WP_032722038	*TrpB*	1.4	WP_128737986	*TrpB*	4.2
WP_032722040	*TrpE*	−0.3	WP_032722040	*TrpE*	3.7
WP_003233236	*TrpP*	−0.1	WP_003233236	*TrpP*	3.5
WP_003230608	*TrpA*	1.1	WP_124058510	*TrpA*	1.9
WP_029725858	*PatB*	−1.4	WP_153256127	*DhaS*	0.8
			WP_024571520	*PatB*	2.5
			WP_021076225	*iaaH*	−1.0
*Terpenoids*					
AGA20733	*IspF*	−1.2	WP_181219684	*fni*	1.8
AGA24047	*Dxr*	−2.0	AGA20733	*IspF*	1.3
WP_003235019	*IspD*	0.0	AGA24047	*Dxr*	−1.1
WP_032722383	*fni*	3.8	WP_003235520	*IspD*	0.3
*2,3-BD*					
6IE0-A	* R-BDH *	2.3	WP_029946299	*bdhA*	−2.1
Serine protease					
WP_014479598	*Isp*	4.8	WP_024572446	*AprX*	5.6
WP_032721588	*AprX*	−1.0	WP_014479598	*Isp*	4.0
WP_015250812	*HtrC*	−1.5	WP_015250812	*HtrC*	1.3
WP_032722717	*Vpr*	3.1	WP_134982250	*Vpr*	4.9
			WP_015382840	*TLS*	3.6
Bacillibactin					
WP_019712937	*DhbC*	−0.8	WP_014480725	*DhbA*	3.1
WP_019712934	*DhbF*	0.0	WP_029946202	*DhbE*	1.6
			WP_106073425	*DhbB*	1.5
			WP_042974556	*DhbC*	2.6
			WP_185183915	*DhbF*	2.5
			WP_129092200	*Btr*	3.9
			KAF1340485.1	*DhbA*	3.9
EPS					
WP_194395382	*EpsB*	0.0	WP_194395382	*EpsB*	2.9
WP_032722561	*EpsE*	−1.3	WP_181220166	*EpsC*	3.2
WP_015714749	*EpsG*	−1.2	WP_128993438	*EpsE*	3.0
WP_003246541	*EpsK*	1.1	WP_015714749	*EpsG*	2.8
			WP_003234384	*EpsK*	3.2
			WP_124059006	*pdeH*	1.6
			WP_166443901	*sugtrans*	4.2
			WP_123373775	*EpsI*	1.7
Spermidine					
WP_003227543	*speE*	−0.2	WP_003227543	*speE*	1.1

FC represents the fold change in expression levels during the stationary phase as compared to the log phase.

## Data Availability

Not applicable.

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
