# Peer review of "Examining the Transcriptomic and Biochemical Signatures of Bacillus subtilis Strains: Impacts on Plant Growth and Abiotic Stress Tolerance"

_ijms, 2023, doi:10.3390/ijms241813720_

Round 1
Reviewer 1 Report
I enjoyed the work, however there are some aspects that should be improved in order to ensure reproducibility. Moreover, some assumptions go very far, and it's better to moderate. Please fin here some comments:
Title:
-I suggest reformulating because the transcriptomic and biochemical detection is more accurate approach instead metabolite spectrum (only checked some key compounds is not making a metabolomic analysis)
Abstract:
- Some aspects are not very clear, jumping from bacteria transcriptome (in which condition?) to their effect in plant transcriptome. I understood the main of the work here, but seems a little ambiguous or not focused topics merging
Introduction:
- Line 34: The comma makes a change in the meaning of the phrase, be careful
- The connection about topics is kind of sharp, jumping between concepts that not always are clearly connected (IAA to bacillibactin, to ACC...)
- Some topics included here are not completely aligned with the aim of the paper, dispersing the attention, I would recommend a full review of this section to try a more concise and focused introduction. Also, a better balance among topics is necessary to get back the results claimed later on.
Materials and Methods:
- Culture conditions? Medium?
- Fungal pathogen Foc TR4, full name is required the first time is mentioned
- Co-culturing and treatment with Arabidopsis should be further described
- Stress test are pots? Is not well described
- Section 4.3 title has a mistake
- Regular PCR conditions are not described
- Statistical software version is required
- Replica of each test is not well addressed
Results:
- Section name in 2.1 is not correct (same as in 2.2)
- Figure 1 has not well referenced the panel C.
- Figure 6 B (pictures) are not clear or representative enough. Plant pictures should be clearer, and use same number of replica per picture panel
- Figure 7 B, pictures has not quality enough and the claimed differences are far from being obvious
- Figure 8 A, quality is very low to differentiate anything. Different scale, different number of replica included....
Discussion:
- The root exudates and profiling are not addressed in this paper to be such an important part of the discussion
- Origin of the strains is not well addressed in order to understand why they behave differently
- The different origin of the strains show to reflect in a different behave in strains. Ok, but what is the insight of this, seems like a vague descriptive reflection. What could be the real implications of the origin in the adaptation mechanisms of the strains is more interesting to discuss
Conclusions:
- Species names should be in italics
- EPS-ABA assumptions are too weak to be conclusive enough
- The different origin of the strains show to reflect in a different behave in strains. Ok, but what is the insight of this, seems like a vague descriptive reflection. What could be the real implications of the origin in the adaptation mechanisms of the strains is more interesting
Author Response
Title:
-I suggest reformulating because the transcriptomic and biochemical detection is more accurate approach instead metabolite spectrum (only checked some key compounds is not making a metabolomic analysis)
Response:
The title has been changed to “Examining the Transcriptomic and Biochemical Signatures of Bacillus subtilis Strains: Impacts on Plant Growth and Abiotic Stress Tolerance”
Abstract:
- Some aspects are not very clear, jumping from bacteria transcriptome (in which condition?) to their effect in plant transcriptome. I understood the main of the work here, but seems a little ambiguous or not focused topics merging
Response:
Line 20-24 in revised manuscript was rewritten.
“qPCR analysis was used to measure transcriptional changes in Arabidopsis seedlings in response to BsCP1 and BsPG1 treatment. The results showed that both bacterial strains had a similar impact on the expression of genes involved in the salicylic acid (SA) and jasmonic acid (JA) signaling pathways. Likewise, genes associated with stress response, root development, and disease resistance showed comparable responses to both bacterial strains.”
Introduction:
- Line 34: The comma makes a change in the meaning of the phrase, be careful
Response:
Line 34 (line 36 in the revised manuscript)
Revised (line 35-37): Rhizobacteria produce diverse metabolites to function as communication signals within the microbial ecosystem. This may provide them an advantage in the competition for resources and habitats against other organisms.
- The connection about topics is kind of sharp, jumping between concepts that not always are clearly connected (IAA to bacillibactin, to ACC...)
- Some topics included here are not completely aligned with the aim of the paper, dispersing the attention, I would recommend a full review of this section to try a more concise and focused introduction. Also, a better balance among topics is necessary to get back the results claimed later on.
Response:
The following content has been added to the introduction to connect various topics.
- “Certain metabolites from PGPR strains can have a direct effect on plant development. For example,” was added to line 37-38.
- “ However, metabolites produced by PGPR strains can also boost plant growth through indirect processes. For example,” was added to line 46-47.
- Metabolites produced by PGPR can promote plant growth by altering multiple plant signaling pathways that are involved in the regulation of adaptive responses under stressful conditions [15]. From line 57- 58 in revised manuscript.
Materials and Methods:
- Culture conditions? Medium?
Response:
“incubated at 30oC for overnight.” was added to line 496 in revised manuscript.
“that had been cultured overnight in LB medium” was added to line 523 in revised manuscript.
- Fungal pathogen Foc TR4, full name is required the first time is mentioned
Response:
“Fusarium oxysporum f. sp. cubense tropical race 4” has been added to line 524 in revised manuscript.
- Co-culturing and treatment with Arabidopsis should be further described
Response:
Line 566-568 in revised manuscript was rewritten to “Four-day-old seedlings of Arabidopsis thaliana (Columbia ecotype) obtained from The Arabidopsis Information Resource (TAIR) were co-cultured with the tested bacterial colonies. These colonies were positioned 4 cm away from the seedlings in 1/2 Murashige and Skoog (MS) medium and then incubated at 23°C under a 16-hour lighting condition.”
- Stress test are pots? Is not well described
Response:
“grown in seedling trays” has been added to line 593 in revised manuscript.
- Section 4.3 title has a mistake
Response:
Title of section 4.3 has been corrected. “Analysis of biochemical properties of isolated bacterial strains”
- Regular PCR conditions are not described
Response:
PCR condition was added to line 490-492 “PCR amplification began with an initial denaturation at 95°C for 5 minutes, followed by 30 cycles of 95°C for 1 minute, an-nealing at 52°C for 30 seconds, and extension at 72°C for 5 minutes. This was concluded with a final extension at 72°C for 3 minutes.”
- Statistical software version is required
Response:
“3.8” was added to line 630 in revised manuscript.
- Replica of each test is not well addressed
Response:
Replica of each test has been added to line 588, 601, 609, 619.
Results:
- Section name in 2.1 is not correct (same as in 2.2)
Response:
2.1 title has been corrected: “Molecular identification of BsCP1 and BsPG1”
- Figure 1 has not well referenced the panel C.
Response:
Locus and link for the genes listed in Figure 1 panel C have been added to supplemental Table 1.
- Figure 6 B (pictures) are not clear or representative enough. Plant pictures should be clearer, and use same number of replica per picture panel
Response:
Figure 6B has been revised.
- Figure 7 B, pictures has not quality enough and the claimed differences are far from being obvious
Response:
Figure 7B has been revised.
- Figure 8 A, quality is very low to differentiate anything. Different scale, different number of replica included....
Response:
Figure 8A has been revised.
Discussion:
- The root exudates and profiling are not addressed in this paper to be such an important part of the discussion
Response:
“Studies have shown that root exudates secreted from different plant species are able to attract various microorganisms to colonize their rhizosphere [34]. Furthermore, rhizobacteria from different environments have different abilities to produce bioactive metabolites that affect plant growth [35].” was added to introduction from line 94 – 96 in revised manuscript.
- Origin of the strains is not well addressed in order to understand why they behave differently
Response:
The origin of the strains is discussed in Section 4.1.
- The different origin of the strains show to reflect in a different behave in strains. Ok, but what is the insight of this, seems like a vague descriptive reflection. What could be the real implications of the origin in the adaptation mechanisms of the strains is more interesting to discuss
Response:
The most significant variance in transcriptional levels and biochemical characteristics between BsCP1 and BsPG1 is linked to exopolysaccharide production. A discussion of the potential reason for this divergence can be found in the section of Discussion from lines 451-459 in revised manuscript.
Conclusions:
- Species names should be in italics
Response:
Corrected.
- EPS-ABA assumptions are too weak to be conclusive enough
Response:
Conclusion from line 638 – 640 has been rewritten.
“Our results indicated that B. subtilis strains with a greater ability to produce EPS showed a stronger capacity to enhance plant tolerance to drought, heat, and copper stress. Additionally, strains that generate larger quantities of EPS are capable of triggering the ABA signaling pathway in Arabidopsis seedlings.”
- The different origin of the strains show to reflect in a different behave in strains. Ok, but what is the insight of this, seems like a vague descriptive reflection. What could be the real implications of the origin in the adaptation mechanisms of the strains is more interesting
Response:
The most significant variance in transcriptional levels and biochemical characteristics between BsCP1 and BsPG1 is linked to exopolysaccharide production. A discussion of the potential reason for this divergence can be found in the section of Discussion from lines 451-459 in revised manuscript.
Reviewer 2 Report
The manuscript by P. E. Chang and co-autors titled “Exploring the Beneficial Metabolite Spectrum of Bacillus subtilis Strains: Impacts on Plant Growth and Abiotic Stress Tolerance” had the purpose to study the influence of two B. subtilis strains isolated from different sources on plant growing in stress conditions. The subject of the research is important, both due to the climatic changes and environmental pollution. Manuscript is interesting, the research results are practical and can be used in plant production, especially in the case of plants growing under conditions of abiotic stress. The methods of the experiment are completely and accurately described and statistical analyses are clear. In the method chapter, please state clearly how many plants constituted one replication and how many replications there were in the experiment. The chapter of results and discussion is successful and the references are adequate. The experimental results are sufficient to justify the conclusions. The obtained results improve knowledge of the mechanisms that cause the B. subtilis strain to increase plant tolerance to abiotic stress.
In my opinion the paper is acceptable for publication in International Journal of Molecular Sciences after necessary complementation.
Author Response
Response:
“Twenty seedlings were included in each treatment.” was added to line 573 - 574 in revised manuscript.
“Experiments were conducted three times.” was added to line 588 – 589 in revised manuscript.
“Forty seedlings were included in each treatment.” was added to line 595 in revised manuscript.
“The drought stress tolerance analysis was performed in triplicate.” was added to line 601 – 602 in revised manuscript.
“Each treatment consisted of 40 seedlings.” was added to line 606 in revised manuscript.
“The heat stress tolerance analysis was performed in triplicate.” was added to line 608 – 609 in revised manuscript.
“Each treatment consisted of 30 seedlings.” was added to line 614 in revised manuscript.
“The analysis for copper stress tolerance was conducted three times.” was added to line 619 in revised manuscript.
Reviewer 3 Report
Review on the manuscript “Exploring the Beneficial Metabolite Spectrum of Bacillussub tilis Strains: Impacts on Plant Growth and Abiotic Stress Tolerance”
The research topic is relevant. The authors studied the transcription profiles for genes associated with the synthesis of antibiotics, the siderophore metabolite bacillibactin, and the EPS biofilm component in two strains of B. subtilis. It was shown that the treatment of plant strains with these data caused the activation of genes associated with the SA and JA signaling pathways, abiotic and biotic responses to stress, as well as with the growth and development of roots. Changes in two antioxidant enzyme (APX and POD) activity and H2O2 and lignin contents were also shown in plants treated with both strains.
In addition, stress experiments (drought, 20 min heating, copper treatment) were carried out on Arabidopsis plants treated with both strains. To confirm the positive effect of both strains on resistance to these effects, the parameters fresh weight and starch content (5-day drought and 5-day recovery period, 20 min. heating) and relative growth inhibition and relative chlorophyll content inhibition (?) after 6-day (?) CuSO4 treatment.
It is not clear why the authors did not use the same set of factors (antioxidant enzyme (APX and POD) activity and H2O2 content) to confirm plant resistance to stress as in the treatment of plants with B. subtilis strains. Which would be logical and allowed the authors to draw more confident conclusions about the effect of these parameters on plant resistance.
Why was starch content used to confirm plant resistance to drought and heat, but not relative chlorophyll content inhibition (as at CuSO4 treatment) ? Chlorophyll is very sensitive to these types of stress.
In addition, in experiments on the effects of stress factors on plants, no analysis of gene expression was carried out, as was done on plants treated with strains
Abstract
26-27 lines – “Apart from the SA and JA signals, the ABA signal triggered by PGPR (“plant growth-promoting rhizobacteria” should be added) bacterial strains could play a crucial role in building long-lasting resistance to various abiotic stresses in the plants they colonize.” - Resistance to 5-day drought/5-day recovery, 20 min. of heating and 6-day (?) CuSO4 treatment is not long-lasting resistance.
Figure 2. – decoding of the abbreviations in the figure should be given
Figure 4 G. – the authors should indicate B. subtilis strains
Figure 5. …lignin content (C), H2O2 content (D), APX activity (E), and POD activity (F) – in roots or leaves?
Figure 8 - The relative chlorophyll content – do the authors mean a decrease in the content of chlorophyll?
Figure 9. Arabidopsis gene expression altered by BsCP1 and BsPG1. – where? In leaves? Root development: nitrilase 2 (NIT2) and NADPH oxidase/respiratory burst oxidase homolog C (RBOHC). – is it also in the leaves?
Discussion
461-480 lines– it's mostly unreasonable and speculative
463-464 lines: “A study has indicated a strong correlation between APX1 expression and plant tolerance to drought, heat stress, and heavy metals [84, 85]”. - this usually means expression during stress or after stress. In [84] “Cytosolic ascorbate peroxidase 1 (APX1) protein and mRNA accumulated during the stress combination.” In [85] “APX1 gene knockout results in Pb tolerance in Arabidopsis. - not in Cu tolerance / during the stress.
472-473 lines: “The ABA signal is an inducer for the expression of starch synthase I [88].Exogenous ABA has been reported to induce the expression of genes involved in starch synthesis, leading to increased starch accumulation in grapevine cuttings [89]”. In [88] – study of maize (Zea mays L.) endosperm, in [89] - study of grapevine cuttings. The authors did not find the studies with such effects on Arabidopsis leaves?
476 line: “P5CS1 is involved in the synthesis of the osmolyte proline, and this metabolic pathway is regulated by ABA [90].” In [90] only in Abstract there is one sentence about putative regulation of P5CS genes by ABA: “Transcription of the P5CS genes is differentially regulated by drought, salinity and abscisic acid, suggesting that these genes play specific roles in the control of proline biosynthesis.” Aim of the study: “Here we describe the genetic characterization of p5cs insertion mutants...” - no study of P5CS1 regulation by ABA in this reference.
This part of the discussion needs serious revision. Authors should avoid discussing what they themselves did not do (did not define). It is not correct to compare changes in enzyme activity and gene expression before stress treatment with literature data on their activity and expression during stress.
In addition, sources for discussion should be chosen more carefully and seriously.
If the authors on stressed plants (by drought, heat, CuSO4) analyze APX and POD activity and H2O2 contents, as well as the expression of the same set of genes as when treated with B. subtilis strains, this will greatly strengthen the article.
Materials and Methods
4.6. qPCR analysis for Arabidopsis gene expression - in roots or leaves?
Supplementary Table 1. – the authors should add Locus for each gene and add a link to the database
Author Response
It is not clear why the authors did not use the same set of factors (antioxidant enzyme (APX and POD) activity and H2O2 content) to confirm plant resistance to stress as in the treatment of plants with B. subtilis strains. Which would be logical and allowed the authors to draw more confident conclusions about the effect of these parameters on plant resistance.
Response:
Water stress is a major contributor to damage in conditions of both drought and heat stress, as evidenced by Takahashi et al. (2020) and Zhao et al. (2020). As a measure of plant resistance against these types of stress, we assessed the ratio of seedlings that survived—specifically those that remained unwilted—following exposure to stress.
Reference:
Takahashi F, Kuromori T, Urano K, Yamaguchi-Shinozaki K, Shinozaki K: Drought stress responses and resistance in plants: From cellular responses to long-distance intercellular communication. Front Plant Sci 2020, 11:1407.
Zhao J, Lu Z, Wang L, Jin B: Plant responses to heat stress: physiology, transcription, noncoding RNAs, and epigenetics. Int J Mol Sci 2020, 22:117.
Why was starch content used to confirm plant resistance to drought and heat, but not relative chlorophyll content inhibition (as at CuSO4 treatment) ? Chlorophyll is very sensitive to these types of stress.
Response:
Water stress plays a significant role in causing harm under both drought and heat stress conditions. To assess the ability of plants to withstand such stress conditions, we looked at the percentage of seedlings that remained resilient after exposure to these adverse conditions. Starch content was used to evaluate plant growth vigor in the post-stress period.
In addition, in experiments on the effects of stress factors on plants, no analysis of gene expression was carried out, as was done on plants treated with strains
Response:
The purpose of examining gene expression was to determine whether bacterial treatments could activate plant cellular pathways that strengthen plant cells' resistance to stress. Since we observed a positive correlation between gene expression assessed prior to stress exposure and the stress-resistance phenotype following stress exposure, we did not conduct further gene expression experiments in the post-stress stage.
Abstract
26-27 lines – “Apart from the SA and JA signals, the ABA signal triggered by PGPR (“plant growth-promoting rhizobacteria” should be added) bacterial strains could play a crucial role in building long-lasting resistance to various abiotic stresses in the plants they colonize.” - Resistance to 5-day drought/5-day recovery, 20 min. of heating and 6-day (?) CuSO4 treatment is not long-lasting resistance.
Response:
In revised manuscript, Line 27-29 was rewritten to “Apart from the SA and JA signaling pathways, ABA signaling triggered by PGPR bacterial strains could play a crucial role in building an effective resistance to various abiotic stresses in the plants they colonize.”
Figure 2. – decoding of the abbreviations in the figure should be given
Response:
“PS: phosphate solubilization. VOCs: volatile organic compounds” was added to the legend of Figure 2 (line 267 in revised manuscript).
Figure 4 G. – the authors should indicate B. subtilis strains
Response:
Figure 4G was revised. BsCP1 and BsPG1 was added to Figure 4G
Figure 5. …lignin content (C), H2O2 content (D), APX activity (E), and POD activity (F) – in roots or leaves?
Response:
Leaf tissues were used for detecting lignin content (C), H2O2 content (D), APX activity (E), and POD activity (F), which was written in section 4.4.
Figure 8 - The relative chlorophyll content – do the authors mean a decrease in the content of chlorophyll?
Response:
Figure 8B was corrected to “relative chlorophyll reduction”.
Figure 9. Arabidopsis gene expression altered by BsCP1 and BsPG1. – where? In leaves? Root development: nitrilase 2 (NIT2) and NADPH oxidase/respiratory burst oxidase homolog C (RBOHC). – is it also in the leaves?
Response:
“the leaf and root tissues” was added to section 4.6 (line 622 in revised manuscript).
Discussion
461-480 lines– it's mostly unreasonable and speculative
463-464 lines: “A study has indicated a strong correlation between APX1 expression and plant tolerance to drought, heat stress, and heavy metals [84, 85]”. - this usually means expression during stress or after stress. In [84] “Cytosolic ascorbate peroxidase 1 (APX1) protein and mRNA accumulated during the stress combination.” In [85] “APX1 gene knockout results in Pb tolerance in Arabidopsis. - not in Cu tolerance / during the stress.
Response:
- Line 469-470 has been rewritten to “The critical function of APX1 within the regulatory network linked to oxidative stress has been established [57].”
- “Increased oxidative damage is a common outcome resulting from drought, heat, and copper stress [16-18].” was added to line 471-472 in revised manuscript.
472-473 lines: “The ABA signal is an inducer for the expression of starch synthase I [88]. Exogenous ABA has been reported to induce the expression of genes involved in starch synthesis, leading to increased starch accumulation in grapevine cuttings [89]”. In [88] – study of maize (Zea mays L.) endosperm, in [89] - study of grapevine cuttings. The authors did not find the studies with such effects on Arabidopsis leaves?
Response:
“Elevated levels of starch have been detected in Arabidopsis leaf tissues when exposed to temperature and osmotic stress conditions [87-89]. The ABA signal is a positive regulator of starch metabolism in maize and rice [90, 91]. In Arabidopsis, the expression of ADP-glucose pyrophosphorylase (AGPase), a key enzyme involved in starch synthesis, is stimulated by sucrose [92]. Furthermore, ABA has the ability to enhance starch synthesis driven by sucrose [93].” was added to line 478-481 in revised manuscript.
476 line: “P5CS1 is involved in the synthesis of the osmolyte proline, and this metabolic pathway is regulated by ABA [90].” In [90] only in Abstract there is one sentence about putative regulation of P5CS genes by ABA: “Transcription of the P5CS genes is differentially regulated by drought, salinity and abscisic acid, suggesting that these genes play specific roles in the control of proline biosynthesis.” Aim of the study: “Here we describe the genetic characterization of p5cs insertion mutants...” - no study of P5CS1 regulation by ABA in this reference.
This part of the discussion needs serious revision. Authors should avoid discussing what they themselves did not do (did not define). It is not correct to compare changes in enzyme activity and gene expression before stress treatment with literature data on their activity and expression during stress.
In addition, sources for discussion should be chosen more carefully and seriously.
Response:
“The P5CS transcript is induced by drought and salinity stress, and the presence of ABA. Additionally, P5CS gene activation in seedlings under salt stress is negated in the ABA-deficient mutant [96].” was added to line 485-487.
If the authors on stressed plants (by drought, heat, CuSO4) analyze APX and POD activity and H2O2 contents, as well as the expression of the same set of genes as when treated with B. subtilis strains, this will greatly strengthen the article.
Response:
qPCR analysis demonstrated that B. subtilis strains induced expression of genes involved in cellular pathways regulating abiotic stress tolerance. Consistently, treatments of B. subtilis strains gained positive results regarding increased abiotic stress tolerance. Therefore, further experiments related to antioxidant enzyme activity and gene expression were not performed.
Materials and Methods
4.6. qPCR analysis for Arabidopsis gene expression - in roots or leaves?
Response:
Sentence from line 622-623 was rewritten to “Total RNA was extracted from the leaf and root tissues of 10-day-old Arabidopsis seedlings cocultured with the tested bacterial strains for 7 days, using the methods described by Lee Downing et al. [116]”
Supplementary Table 1. – the authors should add Locus for each gene and add a link to the database
Response:
Locus and link has been added to Supplementary Table 1
Reviewer 4 Report
In the article, the Authors discuss the still valid issue of the identification and characterization of PGPR strains and their impact on plants.
General comments:
The work lacked a research hypothesis and clearly edited goals of the work. Instead of goals, the authors summarized the results (verses 90-96, p. 3).
The manuscript should be checked for editorial purposes. Sentences begin with a lowercase letter in several places (e.g. lines 388, 392).
In Conclusion, add information which of the tested strains would be better as a potential ingredient of biopreparations.
The References section requires a thorough review.
Detailed comments:
1. Lanes: 26-28, p. 1 - Last sentence of the Abstract, please reword, replace the word "signal".
2. Figure 1- in Table (C) please write “primer” in lower case; use italics in the names of genes and bacterial strains.
In the Table, specify from which base the gene accession number comes; next to the words "length, identity" add the word "sequence".
The caption to Fig. 1 lacks reference to Table (C).
3. Table 1 – write the names of genes in italics
4. Lanes: 430, 446, 494, 559, 562, 564, 567, 570, 581, 598, 603 – year remove (2021, 2022, 2008, 2017, 1987, 2002, 1981, 1984, 2009, 2022, 1992) before number of reference
5. Lanes: 453 and 474 – in words „Bacillus”, „Arabidopsis” insert italics
6. Lane 552 – specify the origin of Arabidopsis seeds
7. Lane 552 - give the full Latin name of Arabidopsis
8. In Materials and methods, specify the volume of inoculation of the plants
9. Provide information whether the RT-PCR product was gel-tested.
10. There is no information about the amount of cDNA used for qPCR
11. Lane 609 - remove the dot in the chapter title
12. Lane 614 - put italics in the name of the bacteria
13. Supplementary Materials - please mark sequence orientations - 5'-3'
14. In Author Contributions - use abbreviations of names and surnames instead of full names
15. References - line 704, - remove the dot
16. Lane 707 - insert italics
17. Lane 709 – remove blank line
18. Lane 718-719 - use journal abbreviation
19. Lane 731 - use the abbreviation
Author Response
- Lanes: 26-28, p. 1 - Last sentence of the Abstract, please reword, replace the word "signal".
Response:
Revision: line 27-29 “Apart from the SA and JA signaling pathways, ABA signaling triggered by PGPR bacterial strains could play a crucial role in building an effective resistance to various abiotic stresses in the plants they colonize.”
- Figure 1- in Table (C) please write “primer” in lower case; use italics in the names of genes and bacterial strains.
In the Table, specify from which base the gene accession number comes; next to the words "length, identity" add the word "sequence".
The caption to Fig. 1 lacks reference to Table (C).
Response:
Figure 1C has been corrected.
- Table 1 – write the names of genes in italics
Response:
Table 1 was corrected.
- Lanes: 430, 446, 494, 559, 562, 564, 567, 570, 581, 598, 603 – year remove (2021, 2022, 2008, 2017, 1987, 2002, 1981, 1984, 2009, 2022, 1992) before number of reference
Response:
Delete year
- Lanes: 453 and 474 – in words „Bacillus”, „Arabidopsis” insert italics
Response:
corrected
- Lane 552 – specify the origin of Arabidopsis seeds
Response:
“obtained from The Arabidopsis Information Resource (TAIR)” was added.
- Lane 552 - give the full Latin name of Arabidopsis
Response:
Full name “Arabidopsis thaliana” was written in section 4.4.
- In Materials and methods, specify the volume of inoculation of the plants
Response:
Line 571-572 was rewritten to “The bacterial solution was administered through foliar spraying on 2-week-old Arabidopsis seedlings once a week for three successive weeks.”
- Provide information whether the RT-PCR product was gel-tested.
Response:
No RT-PCR was conducted in this study.
- There is no information about the amount of cDNA used for qPCR
Response:
In section 4.6, “cDNA was prepared from 2 μg of total RNA”
- Lane 609 - remove the dot in the chapter title
Response:
Could not find a dot in the chapter title in line 609 (revised manuscript line 629).
- Lane 614 - put italics in the name of the bacteria
Response:
Corrected.
- Supplementary Materials - please mark sequence orientations - 5'-3'
Response:
5'-3' added to primer sequences
- In Author Contributions - use abbreviations of names and surnames instead of full names
Response:
Corrected.
- References - line 704, - remove the dot
Response:
Corrected.
- Lane 707 - insert italics
Response:
Corrected.
- Lane 709 – remove blank line
Response:
Corrected.
- Lane 718-719 - use journal abbreviation
Response:
Corrected.
- Lane 731 - use the abbreviation
Response:
Corrected.
Round 2
Reviewer 3 Report
2) Review on the manuscript “Exploring the Beneficial Metabolite Spectrum of Bacillussub tilis Strains: Impacts on Plant Growth and Abiotic Stress Tolerance”
The authors responded to technical comments and made changes to the text.
However, there are some remarks.
Figure captions should be self-sufficient, so the authors should add to the caption to Figure 5 that “… lignin content (C), H2O2 content (D), APX activity (E), and POD activity (F) in the leaves”, and to Figure. 9 which genes were studied in leaves and which ones in roots.
The last sentence of the Discussion (lines 489-490) (“This, in turn, could enhance resistance to stress by minimizing chlorophyll degradation and augmenting the accumulation of starch and proline when subjected to drought, heat, and copper stress.”) does not correspond to the specific results of this study and should be rephrased.
According to the results, since chlorophyll degradation has only been studied after copper exposure (fig.8). Whereas “Starch content was used to evaluate plant growth vigor in the post-stress period” (quote from the authors' answers), that is, not when subjected to drought, heat, but after a recovery period (fig. 6 and 7). These are different mechanisms!
Supplemental Table 1. - if the authors used only two databases, it is enough to indicate them in the notes to the table, and not opposite each gene.
Author Response
Response to Reviewer 3
2) Review on the manuscript “Exploring the Beneficial Metabolite Spectrum of Bacillussub tilis Strains: Impacts on Plant Growth and Abiotic Stress Tolerance”
The authors responded to technical comments and made changes to the text.
However, there are some remarks.
Figure captions should be self-sufficient, so the authors should add to the caption to Figure 5 that “… lignin content (C), H2O2 content (D), APX activity (E), and POD activity (F) in the leaves”, and to Figure. 9 which genes were studied in leaves and which ones in roots.
Response:
“in the leaves” was added to Figure 5 caption.
“Changes in gene expression were detected by qPCR analysis using total RNA prepared from whole seedlings treated with BsCP1 and BsPG1” was added to Figure 9 caption.
The last sentence of the Discussion (lines 489-490) (“This, in turn, could enhance resistance to stress by minimizing chlorophyll degradation and augmenting the accumulation of starch and proline when subjected to drought, heat, and copper stress.”) does not correspond to the specific results of this study and should be rephrased.
Response:
Rephrase: The superior EPS production by BsPG1 could stimulate the ABA signaling pathway, which could enhance resistance to stress by minimizing chlorophyll degradation and augmenting the accumulation of starch and proline when subjected to drought, heat, and copper stress.
According to the results, since chlorophyll degradation has only been studied after copper exposure (fig.8). Whereas “Starch content was used to evaluate plant growth vigor in the post-stress period” (quote from the authors' answers), that is, not when subjected to drought, heat, but after a recovery period (fig. 6 and 7). These are different mechanisms!
Response:
Water stress is a significant factor causing damage under both drought and heat stress conditions, but not under copper stress. Therefore, the assays for detecting drought and heat stress tolerance were different from those detecting copper stress.
Supplemental Table 1. - if the authors used only two databases, it is enough to indicate them in the notes to the table, and not opposite each gene.
Response:
Corrected.